# Construction and Textural Properties of Plant-Based Fat Analogues Based on a Soy Protein Isolate/Sodium Alginate Complex Coacervation System

**DOI:** 10.3390/foods14244355

**Published:** 2025-12-18

**Authors:** Yilin Tu, Guijiang Liang, Zhaojun Wang, Maomao Zeng, Zhiyong He, Qiuming Chen, Jie Chen

**Affiliations:** 1State Key Laboratory of Food Science and Resources, Jiangnan University, Wuxi 214122, China; yilintuu@163.com (Y.T.); liangguijiang0103@163.com (G.L.); zhaojun.wang@jiangnan.edu.cn (Z.W.); mmzeng@jiangnan.edu.cn (M.Z.); zyhe@jiangnan.edu.cn (Z.H.); chenqm@jiangnan.edu.cn (Q.C.); 2School of Food Science and Technology, Jiangnan University, Wuxi 214122, China

**Keywords:** fat analogues, microcapsules, complex coacervation, texture

## Abstract

This study focused on the preparation of microcapsules that simulate adipose tissue cells via complex coacervation, followed by the formation of block-like fat analogue products through gelation. The results indicated that microcapsules obtained by encapsulating coconut oil with soy protein isolate (SPI) and sodium alginate (SA) through a complex coacervation process could serve as effective fat substitutes in meat products. When the mass ratio of SPI to SA was 3:1, the core-to-wall mass ratio was 1:1, and the total wall material concentration was 3% (*w*/*v*), the oil loading rate of the microcapsules reached 39.17%. The particle size of the oil-loaded microcapsules was mainly distributed between 40–180 μm, which was comparable to the size of fat cells in animal adipose tissue. Microcapsules (50%, *w*/*w*) were mixed with a 5% (*w*/*v*) curdlan dispersion and heated at 95 °C for 60 min to form fat analogues. The fat analogues demonstrated significantly reduced cooking loss, enhanced textural rigidity, and superior chew resistance, achieving performance metrics comparable to those of natural adipose tissue. This dual-phase strategy—combining interfacial engineering of lipid microcapsules with polysaccharide-mediated gelation—provides a promising approach for developing sustainable, plant-based fat alternatives in meat product reformulation. The methodology not only addresses texture and flavour challenges in fat replacement but also enables precise control over lipid content, supporting applications in healthier food systems.

## 1. Introduction

Excessive consumption of animal meat not only poses risks to human health, increasing the prevalence of obesity, hyperlipidemia, diabetes, and other related diseases [1], but also exerts pressure on the environment due to high market demand, leading to the overexploitation of resources and increased greenhouse gas emissions [2]. Moreover, with growing consumer awareness of healthy diets and body weight management, there is an increasing demand for foods that are low in sugar and fat, high in protein, and yet retain desirable taste and flavour. Consequently, the development of plant-based fat and meat analogues has become an inevitable trend. Fat analogues can simulate the texture characteristics of fat. On the one hand, they help reduce the health risks associated with excessive fat intake, and on the other hand, they provide a sensory experience similar to that of saturated fat, thereby achieving a coordinated balance between reducing saturated fat intake and maintaining food quality such as flavour [3].

Currently, there are three main types of fat analogues used in meat products. The first type is polysaccharide-based hydrocolloids system, this category relies on hydrophilic colloids (e.g., modified starch, carrageenan, gellan gum, konjac glucomannan) forming a three-dimensional hydrated network in the aqueous phase through hydrogen bonding, ionic cross-linking, and polymer chain entanglement. This network physically immobilizes significant amounts of water, and by modulating its rheological properties (e.g., viscoelasticity, yield stress, and shear-thinning behaviour), it simulates the lubricity, juiciness, and soft texture of animal fat [4,5]. Such systems offer the advantage of low energy density, significantly reducing total caloric content by substituting lipids with water. They also exhibit excellent structural stability, demonstrating favourable thermal and freeze–thaw resistance, making them suitable for high-temperature processing and cold-chain logistics. Furthermore, their functional versatility allows synergistic interactions between different hydrocolloids, enabling precise texture customization [6]. However, limitations include flavour deficiency, as they lack lipid-soluble flavour compounds, thus failing to replicate the characteristic aroma and rich taste of animal fats. Nutritionally, they are deficient in essential fatty acids, resulting in incomplete nutrient substitution. Additionally, their “gel-like” mouthfeel may deviate from the typical “melting” sensation of natural fat. The second type is an emulsion gel system; emulsion gels are soft solid materials formed by encapsulating emulsion droplets as active filler particles within a continuous gel matrix [7]. Their fabrication typically involves two stages: First, a high-energy emulsification process (e.g., high-pressure homogenization or ultrasonication) prepares an oil-in-water (O/W) emulsion stabilized by emulsifiers. Common emulsifiers include proteins (e.g., whey protein, soy protein, sodium caseinate), surfactants (e.g., Tween, Span), and polysaccharides (e.g., xanthan gum, acacia gum) [8]. Subsequently, the continuous protein phase is gelled via thermal, acid-induced, or enzymatic cross-linking (e.g., transglutaminase), or the emulsion is incorporated into polysaccharide solutions (e.g., κ-carrageenan, gelatin) for cold-set gelation, ultimately forming a robust gel network [9]. The long-term stability of emulsion gels is closely related to interactions among their components (such as emulsifiers and gelling agents), including electrostatic forces, van der Waals forces, hydrogen bonding, covalent exchange, hydrophobic interactions, and steric hindrance effects. These interactions determine the microstructure, rheological properties, mechanical strength, and long-term stability of emulsion gels [10]. These systems exhibit high structural biomimicry, with microstructures resembling animal adipose tissue, effectively mimicking fat’s texture and oral disintegration behaviour. They also function as multifunctional carriers, serving as fat replacers, flavour vehicles, and delivery systems for bioactive compounds (e.g., fat-soluble vitamins). Their superior physicochemical properties significantly enhance the water-holding capacity, emulsion stability, and texture of low-fat meat products. Limitations include matrix constraints, as traditional reliance on animal-derived proteins (e.g., gelatin, whey protein) restricts their application in strictly plant-based or religiously compliant foods. Moreover, process complexity demands precise control over emulsification and gelation steps to avoid emulsion instability or inadequate gel strength. The third type is oleogel system, also termed “organogels,” represent a strategy to structure liquid vegetable oils into solid or semi-solid states without aqueous phases [11]. The core mechanism involves the self-assembly of low-molecular-weight gelators (e.g., monoglycerides, plant waxes, β-sitosterol + γ-oryzanol, ethyl cellulose) into three-dimensional supramolecular networks (e.g., fibrils, platelets, or nanobelts) within the oil phase. These networks physically immobilize liquid oils via steric hindrance and capillary effects, driven primarily by non-covalent interactions such as van der Waals forces, hydrogen bonding, π–π stacking, and crystalline network synergies [12,13]. Key advantages include customizable lipid composition, enabling direct use of unsaturated fatty acid-rich vegetable oils (e.g., olive oil, flaxseed oil) to fundamentally improve the product’s fatty acid profile. They also achieve complete flavour retention, as the oil matrix inherently carries and releases lipid-derived flavours. Furthermore, their high simulation precision allows exact control over melting curves and solid fat content by adjusting gelator type, concentration, and crystallization temperature to match the sensory attributes of specific animal fats (e.g., lard, tallow) [14]. However, limited gelator availability persists, with few high-efficacy options approved for food use, often at elevated costs. Some gelators (e.g., plant waxes) may impart a waxy mouthfeel or undesirable aftertaste. Additionally, the network’s thermal sensitivity renders it prone to structural collapse (Ostwald ripening) under excessive temperatures [15,16].

The quality of existing fat analogues remains inferior to that of traditional animal fat, largely because they lack a microstructure resembling adipose tissue and insufficiently tolerate external forces and temperature fluctuations [17]. Animal fat tissue is composed of a hydrated connective tissue matrix embedded with adipocytes ranging from 50 to 200 μm in diameter. Adipocytes are predominantly unilocular white fat cells, with a central lipid droplet occupying approximately 90% of the cell volume, while the nucleus is displaced to one side and the cytoplasm forms a thin layer surrounding the droplet [18,19,20]. At a fundamental level, these lipid droplets are simple lipid condensates within the cell’s aqueous environment, enveloped by a phospholipid membrane and separated from other organelles.

This work aimed to prepare oil-loaded microcapsules to mimic animal adipocytes by using the complex coacervating method with coconut oil as the core material and protein and polysaccharide as the wall material. In aqueous solutions, proteins and polysaccharides can interact through electrostatic forces, hydrogen bonding, and hydrophobic interactions to form soluble complexes or coacervates. The oil loading rate, yield of complex condensation and particle size of the oil-loaded microcapsules were investigated. Then, curdlan was used as the gel matrix and mixed with the oil-loaded microcapsules to prepare the fat analogues by high-temperature gelation. The steaming loss rate, mechanical properties and sensory evaluation of the fat analogues were measured and compared with pork and beef fat.

## 2. Materials and Methods

### 2.1. Materials

Soy protein isolate (SPI, SD-100) was purchased from Linyi Shansong Biological Products Co., Ltd. (Jinan, Shandong, China) (content > 90%). The coconut oil was purchased from Coconut Rich Cold Press Co., Ltd. (Haikou, Hainan, China). The curdlan was purchased from Sinopharm Chemical Reagent Co., Ltd. (Shanghai, China). The other reagents used in this experiment were of analytical grade and purchased from Sinopharm Chemical Reagent Co., Ltd. (Shanghai, China). Mili-Q water was used to prepare all the aqueous solutions.

### 2.2. Preparation of Oil-Loaded Microcapsules

The preparation model of fat analogues is shown in Figure 1. on the preparation method of Zhao et al. [21], Hu et al. [22] and Yuan [23], SPI (4.5%, *w*/*v*), SA (1.5%, *w*/*v*), gum arabic (GA, 4%, *w*/*v*) and pectin (0.8%, *w*/*v*) stock solutions were separately prepared by dissolving the sample powders in Milli-Q water under gentle stirring of 800 rpm at 40 °C for 2 h. The pH value of SPI was adjusted to 9.0 with 0.1 M NaOH, then coconut oil was, respectively, mixed with the SPI stock solution at the mass ratios of 7:1, 5:1, 3:1, 2:1, 1:1 and 1:2. Then the SA, GA and pectin stock solutions were separately mixed with the SPI stock solution at the mass of 1:1, 1:3 and 1:6. The total wall material concentration was set as 1% (*w*/*v*), 2% (*w*/*v*), 3% (*w*/*v*) and 4% (*w*/*v*). The total biopolymer concentration was set as 3% (*w*/*v*). The pH of the blends with different protein to polysaccharide mixing ratios was modified by 0.1 M HCl, ranging from 2.5 to 4.0 refer to Faezeh et al. [24]. The resulting blends were held at 40 °C under a stirring rate of 500 rpm for 5 min, 10 min, 20 min and 30 min, respectively. After that, the samples were stored at 4 °C for further analysis.

**Figure 1 foods-14-04355-f001:**
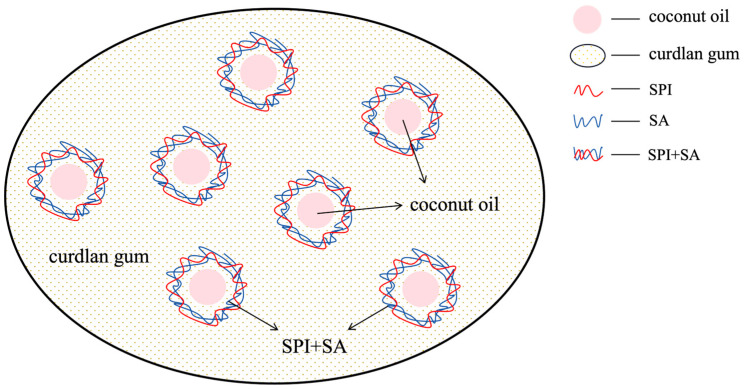
Schematic diagram of the structural principle of microcapsule–gel fat analogues.

### 2.3. Determination of Oil Loading Rate of Oil-Loaded Microcapsules

Determination of oil loading rate of oil-loaded microcapsules has been done as follows [25]: An amount of 200 mg of oil-loaded microcapsules, accurately weighed, was poured into 10 mL of cyclohexane and then ultrasonicated for 1 min to extract coconut oil. After ultrasonication, samples were filtered using a 0.45 μm filter and then the microcapsules were dried at 30 °C for 2 h with the aim of eliminating the residual solvent (cyclohexane). The oil loading rate of oil-loaded microcapsules is determined using the mass differences of microcapsules containing coconut oil and capsules after oil extraction according to Equation (1):
(1)$$oil\;\;loading\;rate\left(\%\right)=\frac{\left(m_{2}-m_{1}\right)}{m_{2}}\times 100$$ where *m*_2_ is the microcapsules mass before ultrasonication, g; *m*_1_ is the microcapsules mass after oil extraction. For each analysis, at least three separate samples were made and measured.

### 2.4. Determination of Condensation Yield of Oil-Loaded Microcapsules

The complex condensation solution was poured into a centrifuge tube and centrifuged at 3000 rpm for 30 min. The white solid at the bottom of the centrifuge tube was the complex, and cleaned three times with deionized water. The supernatant and the repolymer were removed and freeze-dried. The yield of the complex polymer was calculated following Equation (2):
(2)$$Y\;\left(\%\right)=\frac{m}{M}\times 100$$ where *Y* was the yield of complex polymer; m was the mass of the repolymer after drying; *M* was the mass of the total input material before any reaction. For each analysis, at least three separate samples were made and measured.

### 2.5. Preparation of Fat Analogues

The curdlan dispersion liquid was prepared by dissolving the sample powders in Milli-Q water and the pH values of the above prepared dispersion liquid were adjusted to 9.0 with 0.1 M NaOH. Curdlan solutions were formulated at concentrations of 3%, 4%, and 5% (*w*/*v*). Then the collected oil-loaded microcapsules were mixed with curdlan dispersion at the mass ratios of 1:1, 2:1 and 3:1, respectively. The resulting mixture was held at 95 °C for 15 min, 30 min, 45 min and 60 min, respectively.

### 2.6. Steaming Loss of Fat Analogues

The steaming loss of fat analogues and pork fat tissue was measured according to the method of Scheeder et al. [26]. Samples were trimmed to the size of 30 mm × 30 mm × 30 mm for testing. The sample was vacuum-packed and subjected to steaming in a hot water bath (100 °C). The mass of the steamed sample was weighed and recorded as M_2_ and the raw sample was M_1_ to calculate the steaming loss according to Equation (3):
(3)$$steaming\;loss(\%)=\frac{M_{1}-M_{2}}{M_{1}}\times 100$$ where *M*_1_ was the weight of the sample before steaming, g; *M*_2_ was the weight of the sample after steaming, g. For each analysis, at least three separate samples were made and measured.

### 2.7. Mechanical Properties of Fat Analogues

Animal fats and experimentally prepared fat analogues were both subjected to high-temperature treatment at 100 °C for 5 min. The mechanical properties of the fat analogues were evaluated using a texture analyzer (TA) through cutting, puncture, and compression tests. An XT plus texture analyzer (Stable Micro Systems Ltd., London, UK) equipped with Exponent Connect software, Version 6,1,26,0 (Hamilton Texture Technologies, Hamilton, MA, USA) was used.

#### 2.7.1. Texture Profile Analysis (TPA)

A compression test was conducted according to the method refer to Wang et al. and Seyyed et al. [27,28], using a p/50 flat- bottom cylindrical probe for texture profile analysis to simulate the process of human teeth chewing food. The fat tissue analogues and animal fat tissues were cut into a uniform square shaped of 10 mm × 10 mm × 10 mm for testing (*n* = 3 per group). In TPA process, parameters were set as follows: pretest speed, 2.0 mm/s; test speed, 1.0 mm/s; posttest speed, 2.0 mm/s; and compression degree, 40%. Two continuous compressions were performed and TPA texture attributes (hardness, adhesiveness, cohesiveness, springiness, and chewiness) were extracted from the force–time curve. Each experiment was carried out in triplicate.

#### 2.7.2. Puncture Test

A p/2 probe was used for the puncture test. The fat tissue analogues and animal fat tissues were cut into a uniform square shaped of 10 mm × 10 mm × 2 mm for testing (*n* = 3 per group). In puncture test process, parameters were set as follows: calibration height, 20.0 mm, target force, 5.0 N; pretest speed, 1.0 mm/s; test speed, 1.0 mm/s; posttest speed, 5.0 mm/s; and penetration depth, 70%. Each experiment was carried out in triplicate.

#### 2.7.3. Shear Test

A shear blade was used for the cutting test. The fat tissue analogues and animal fat tissues were cut into a uniform square shaped of 10 mm × 10 mm × 2 mm for testing (*n* = 3 per group). In shear test process, parameters were set as follows: calibration height, 20.0 mm; target force, 5.0 N; pretest speed, 1.5 mm/s; test speed, 1.5 mm/s; posttest speed, 10.0 mm/s; and cutting distance, 100%. Each experiment was carried out in triplicate.

### 2.8. Microscopic Structure

The samples were pre-frozen using a cryo-embedding medium and sectioned into 20 μm slices with a cryostat microtome. The sections were then mounted onto glass slides. A Nile Red solution (0.1% *w*/*v*, 30% acetone) and a Fluorescein Isothiocyanate (FITC) solution (0.1% *w*/*v*, 30% acetone) were applied to stain lipids and proteins, respectively. After staining for 10 min in the dark, excess fluorescent dye was removed by washing with 30% acetone. The samples were observed using a TCS SP8 confocal laser scanning microscope (CLSM) (Leica, Wetzlar, Germany) equipped with an HC PL APO 10×/0.4 CS objective. Excitation wavelengths of 552 nm (for Nile Red) and 488 nm (for FITC) were employed. All images were acquired at a resolution of 1024 × 1024 pixels.

### 2.9. Sensory Evaluation Profiles of Fat Analogues

The sensory evaluation profiles followed the method of Wauters et al. [29] with a slight modification. Twelve sensory evaluators (six females and six males, aged 22–32 years old) from Jiangnan University (Wuxi, Jiangsu, China) were selected and trained in accordance with ISO 8586 guidelines (ISO 8586, 2023) [30], participated in two training sessions where they learned to identify and quantify descriptors related to flavour and odour. After the group discussion, semi-trained panellists evaluated the parameters of odour, colour, characteristic flavour, characteristic aroma, juiciness and texture. The evaluators rated the strength of each attribute out of 5 (1: none to 5: very strong). The protocol was approved by the Medical Ethics Committee of Jiangnan University (JNU202406RB005). The results of the sensory evaluation were finally plotted as sensory profile maps.

### 2.10. Statistical Analysis

Texture data were acquired using the Exponent Connect software, and the experimental results are expressed as mean ± standard deviation. Variance and mean values were processed using Microsoft Excel 2021. Significant differences were analyzed using Statistix 9.0 software (General Linear Model and Least Significant Difference method, LSD). Graphs were generated using Excel and GraphPad Prism 9.5. CLSM images were composited using ImageJ1 software.

## 3. Results and Discussion

### 3.1. Preparation of Oil-Loaded Microcapsules

#### 3.1.1. Effects of Wall Material Type, Proportion, Wall Material Concentration and Mass Ratio of Core to Wall on Oil Loading Rate of Oil-Loaded Microcapsules

The oil loading rate of fat analogues significantly influences their flavour, taste, and potential applications [31]. The effects of wall material type, core-to-wall mass ratio, wall material concentration, and wall material composition on the oil loading rate of the microcapsules were systematically investigated. As shown in Figure 2A, when SPI and SA were used as wall materials with a core-to-wall mass ratio of 1:1, the oil loading rate reached its maximum value of 39.17%. Deviations from this ratio—either higher or lower core material content—were detrimental to oil encapsulation. Excessive core material led to wall material cracking, while insufficient core material reduced encapsulation efficiency. Microcapsules using SPI-gum arabic (GA) or SPI-pectin as wall materials exhibited significantly lower oil loading rates than those prepared with SPI-SA. This difference may be attributed to tighter complex coacervation in the SPI-SA system, which facilitated stable encapsulation during stirring and minimized core material leakage. Previous studies also support this observation Mu Et Al. [32] reported that SPI or sodium caseinate combined with polysaccharides can serve as effective microencapsulating agents for green coffee oil, while Li Et Al. [33] demonstrated that moderate SA addition enhances the stability and bioaccessibility of SPI emulsion systems. Figure 2B illustrates the effects of wall material concentration and composition on the oil loading rate when SPI-SA was used as the wall material with a core-to-wall mass ratio of 1:1. The results show that when the total wall material concentration was 3% (*w*/*v*) and the SPI-to-SA mass ratio was 3:1, the positive charges of SPI and negative charges of SA were effectively neutralized. Under these conditions, the condensation yield was maximized, resulting in the highest oil loading rate. Deviations in SPI content—either higher or lower—led to incomplete coacervation and a subsequent decrease in oil loading efficiency.

#### 3.1.2. Effects of pH and Mixing Time on the Yield of Oil-Loaded Microcapsules

During the complex coacervation process, when the pH was significantly below the isoelectric point of the protein, the positively charged soybean protein isolate (SPI) and negatively charged SA attracted each other, resulting in the formation of insoluble complexes through interionic interactions. Concurrently, coconut oil dispersed in the solution was encapsulated to form oil-carrying microcapsules [34]. The isoelectric point of SPI is approximately 4.1 [35]. To ensure effective complex coacervation of the two wall materials, the system pH prior to coacervation should be higher than 4.1. In this study, the pH and stirring time during coacervation were varied to investigate their effects on the condensation yield. Figure 3A illustrated the relationship between pH on the condensation yield during the complex coacervation of SPI and SA. At pH values significantly below the isoelectric point of SPI, SPI carried a strong positive charge, while SA remain negatively charged. This created robust electrostatic attraction between the two biopolymers, leading to efficient complex coacervation and higher condensation yields. As pH approached the pI of SPI (e.g., pH 3.5 and pH 4.0), the net charge of SPI diminished, weakening the electrostatic interactions. This reduced the formation of insoluble SPI-SA complexes, resulting in a decline in condensation yield. The maximum yield at pH 3.0 reflected an optimal balance between SPI protonation (ensuring sufficient positive charge) and SA ionization (maintaining negative charge), maximizing coacervate formation. Faezeh Et Al. [36] investigated the phase separation behaviour of the NaCas/HMP coacervate and its kinetics turbidity and the thermal, rheological and structural behaviour of the coacervates was evaluated at the selected pH values. The results also confirmed that thermal and mechanical stability of the NaCas/HMP coacervates was improved at pH 3.3. Similarly, extending the coacervation time initially enhanced the yield, but prolonged stirring eventually caused a decline. That was because prolonged stirring resulted in the breakdown and dispersal of the polymeric structures. The optimal condition for complex coacervation was determined to be pH 3.0 with continuous stirring for 20 min.

### 3.2. Preparation of Fat Analogues

#### 3.2.1. Effects of the Addition Amount of Oil-Loaded Microcapsules and Concentration of Curdlan on Steaming Loss and Mechanical Properties of Fat Analogues

In this study, the concentration of curdlan and the mass ratio of microcapsules to curdlan were varied to investigate their effects on steaming loss and the mechanical properties of fat analogues, with comparisons made to pork and beef fat. As shown in Figure 4A, the steaming loss of the fat analogues decreased with increasing curdlan concentration. This can be attributed to tighter gel cross-linking at higher curdlan levels, which more effectively retains water. Conversely, increasing the proportion of microcapsules led to higher steaming loss, as the relative decrease in curdlan content reduced the degree of gel cross-linking, facilitating water leakage at high temperatures. Notably, when the curdlan concentration was 5% (*w*/*v*) and the mass ratio of microcapsules to curdlan was 1:1, the steaming loss of the fat analogues was closest to that of pork fat. Figure 4B–D compares the hardness, springiness, and chewiness of the cooked fat analogues and pork fat. Hardness was defined as the peak force required to achieve 40% deformation, while chewiness represents the energy needed to masticate a solid food until ready for swallowing, calculated as the product of hardness, cohesiveness, and springiness [37,38]. The results indicated that reducing the amount of microcapsules added increased the hardness, springiness, and chewiness of the fat analogues. While the springiness of the fat analogues was comparable to cooked pork fat, their hardness and chewiness were significantly lower. Increasing the concentration of curdlan led to notable improvements in all three textural properties, owing to tighter gel cross-linking and enhanced gel strength. However, higher curdlan concentrations and mass ratios also reduced the overall fat content of the analogues. Figure 4E,F presents the shear and puncture test results, reflecting the surface strength and toughness of the samples. Greater toughness corresponds to higher energy required for chewing [39]. As the proportion of microcapsules decreased, the mechanical indices of the fat analogues increased, yet they remained significantly lower than those of pig backfat and beef belly fat, and were closer to those of bacon and pork fat. Based on these results, a curdlan concentration of 5% (*w*/*v*) with a 1:1 mass ratio to oil-loaded microcapsules was identified as the optimal preparation condition.

#### 3.2.2. Effects of Different Heating Time on Steaming Loss and Mechanical Properties of Fat Analogues

When curdlan is heated to high temperatures, its helical molecular structure unwinds. During subsequent cooling, the molecular chains associate through strong hydrogen bonds, forming a stable three-dimensional network that effectively entraps water molecules. This network is robust enough to withstand further thermal processing [40]. In this study, the heating time of the gel was varied to investigate its effect on steaming loss and the mechanical properties of the fat analogues, with comparisons made to pork fat. As shown in Figure 5A, the steaming loss of the fat analogues decreased with increasing heating time. This trend can be attributed to more complete and tighter gel cross-linking with prolonged heating, which reduces water leakage. However, after 45 min of heating, the reduction in steaming loss levelled off, indicating that the gel had largely reached its maximum cross-linking. At this point, the steaming loss of the fat analogues was comparable to that of pork fat, and when heating exceeded 45 min, the cooking loss of the analogues was even lower than that of pork fat. Figure 5B–D show that the hardness and chewiness of the fat analogues increased with longer heating times. This improvement is consistent with more complete and tighter gel cross-linking over extended heating, which enhances gel strength. In contrast, the springiness of the fat analogues showed only minor improvement and no significant differences were observed. Similarly to the steaming loss results, the increase in hardness and chewiness plateaued after 45 min of heating, reflecting the near-complete cross-linking of the gel. Figure 5E,F present the shear and puncture test results, respectively. Although the mechanical indices of the fat analogues increased with longer heating, they remained lower than those of pig backfat and beef belly fat, while approaching the levels observed in pig suet, bacon, and beef pork.

### 3.3. Microscopic Structure

Adipose tissue consists of a collagen network filled with adipocytes containing fat [41,42]. Its unique structure and function are closely related to its fat composition, which is rich in saturated fatty acids (SFAs), and its collagen network, both of which influence the quality attributes of meat products [43]. Figure 6 showed the microstructures of animal fats from different sources and the fat analogues under confocal microscopy. White adipose tissue is a highly organized organ, with a structural core composed of adipocytes containing large unilocular lipid droplets. These cells are interconnected by abundant blood vessels and nerves and supported by a delicate extracellular matrix, collectively forming an efficient system for energy storage and endocrine function. The experimental results demonstrate that confocal microscopy combined with specific fluorescent staining can clearly resolve the spatial distribution and interrelationships of lipid and protein components in multiple colours. The adipose tissue consists of closely packed adipocytes forming a honeycomb-like network, with proteins (green) surrounding lipids (red) and adipocyte sizes ranging from 50 to 250 μm. Animal adipose tissue cells are irregular in size, polygonal in shape, and tightly wrapped by connective tissue. Compared with pork fat, beef fat exhibits a denser and stronger collagen network. As shown in Figure 6F, the microcapsules prepared in this study, formed by encapsulating lipids with proteins and polysaccharides, are generally multinucleated, with an overall lipid content significantly lower than that of animal fat. Compared with the oil content of animal adipose tissue, the fat analogues developed here not only simulate the microstructure of animal adipose tissue but also effectively reduce water and oil loss during cooking, thereby maintaining juiciness and helping control fat intake in fat-containing food products. (Note: due to the inherent brittleness of curdlan, the sectioning and staining process may induce some structural dispersion.)

### 3.4. Photographic Appearance and Sensory Evaluation Profiles of Fat Analogues

This study successfully developed fat analogues that closely resembled pork back fat in both macroscopic morphology and sensory characteristics. As shown in Figure 7A, after high-temperature cooking, the fat analogues exhibited remarkably similar macroscopic morphological features to pork back fat, specifically demonstrating equivalent patterns of surface gloss variation, comparable degrees of yellowness development (attributed to the combined effects of Maillard reactions and lipid oxidation), and analogous fat network restructuring behaviour. These morphological consistencies indicated that the analogues effectively replicated the core physicochemical transformations of pork back fat during thermal processing, including fat melting, protein denaturation, and structural reorganization. Sensory evaluation profiles (Figure 7B) further confirmed the high degree of similarity between the fat analogues and pork back fat in key sensory attributes. Specifically, no significant differences were observed in core indicators including colour, textural hardness, adhesiveness, and characteristic flavour intensity, demonstrating sensory performance comparable to animal fat. Although slight differences in juiciness were noted due to the high water-holding capacity of the polysaccharide-based gel matrix, the analogues achieved a high level of simulation of pork back fat in overall acceptability and key textural parameters. Compared to existing coconut oil-based fat substitutes in the literature—whose triglyceride composition fundamentally differs from animal fats, resulting in suboptimal melting characteristics, crystallization behaviour, and oral breakdown properties [44]—this study successfully achieved stable encapsulation of plant oils and precise textural regulation through the construction of a protein-polysaccharide microcapsule system and composite gel network. This approach effectively overcomes common limitations of traditional plant-based fat analogues in both mechanical properties (such as hardness and adhesiveness) and sensory attributes (such as creaminess and flavour persistence). In summary, the developed fat analogue demonstrates remarkable similarity to pork backfat across multiple dimensions including visual appearance, textural properties, and flavour characteristics, providing an effective solution for the development of plant-based fat replacement systems. Future research will focus on the incorporation of flavour precursors and optimization of oil release profiles to further enhance its sensory quality and overall acceptability.

## 4. Conclusions

This study successfully developed fat analogues using oil microcapsules fabricated through complex coacervation of SPI and SA, embedded within a curdlan-based gel matrix. The optimized microcapsules exhibited favourable encapsulation characteristics, including high oil loading capacity (≈40%) and coacervation yield (91%). When incorporated into curdlan gels at 5% (*w*/*v*) concentration with 50% (*w*/*v*) microcapsule proportion and 45 min heating, the resulting fat analogues demonstrated textural properties and steaming loss comparable to natural pork backfat, along with satisfactory sensory acceptance. These findings validate the feasibility of utilizing microencapsulation technology to create fat analogues with tailored physicochemical and sensory attributes. Furthermore, the proposed strategy shows significant potential for broader applications in other food systems, particularly in dairy and bakery products. In dairy products, the microcapsules could be used to encapsulate probiotics, or lipophilic bioactive compounds, thereby masking off-flavours, enhancing stability, and enabling controlled release. In bakery applications, these microcapsules may serve as effective fat mimetics in items such as biscuits and cakes, imparting desired softness and moisture while reducing fat content. Thus, the proposed system not only offers a novel approach to structuring meat analogues, but also provides a versatile technological platform for texture modulation and functional ingredient delivery in other sectors.

## Figures and Tables

**Figure 2 foods-14-04355-f002:**
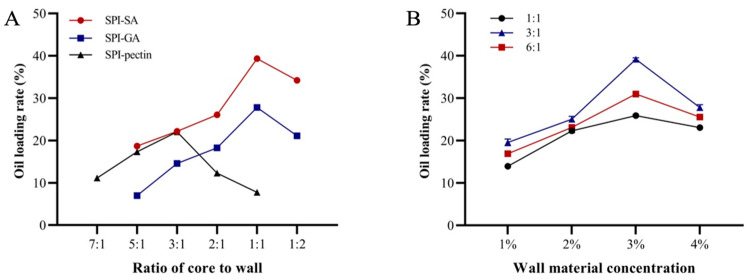
Effects of microcapsule preparation conditions on the oil loading rate of oil-loaded microcapsules. (**A**) Ratio of core to wall; (**B**) wall material concentration.

**Figure 3 foods-14-04355-f003:**
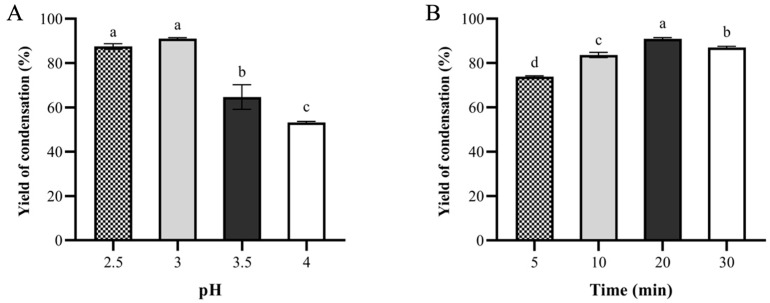
Effects of microcapsule preparation conditions on condensation yield of oil-loaded microcapsules. (**A**) pH; (**B**) Time. Different letters indicate a statistically significant difference (*p* < 0.05).

**Figure 4 foods-14-04355-f004:**
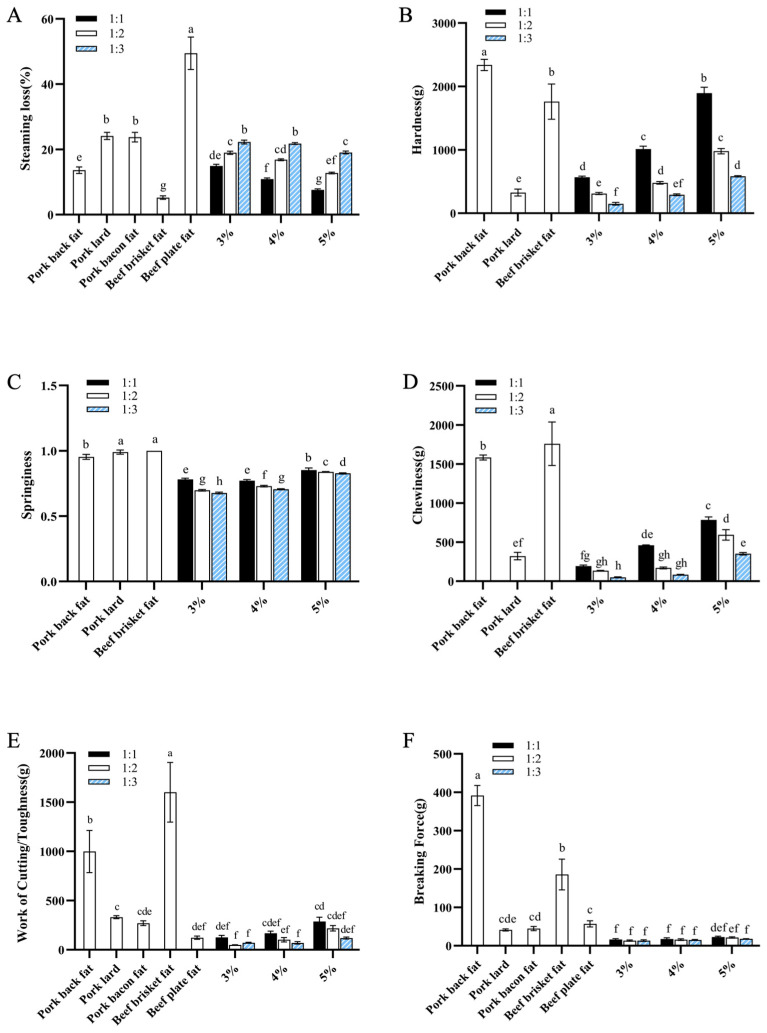
Effects of the addition amount of oil-loaded microcapsules and the concentration of curdlan on the steaming loss (**A**), hardness (**B**), springiness (**C**), chewiness (**D**), work of cutting (**E**) and breaking force (**F**) of fat analogues. Different letters indicate a statistically significant difference (*p* < 0.05).

**Figure 5 foods-14-04355-f005:**
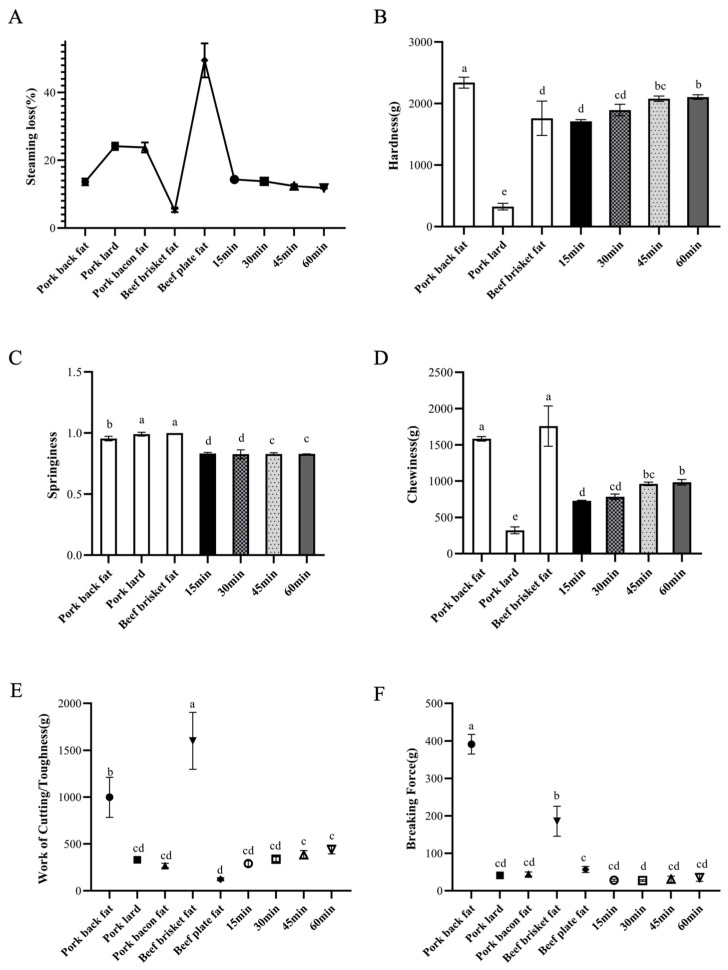
Effects of different heating time on steaming loss (**A**), hardness (**B**), springiness (**C**), chewiness (**D**), work of cutting (**E**) and breaking force (**F**) of fat analogues. Different letters indicate a statistically significant difference (*p* < 0.05).

**Figure 6 foods-14-04355-f006:**
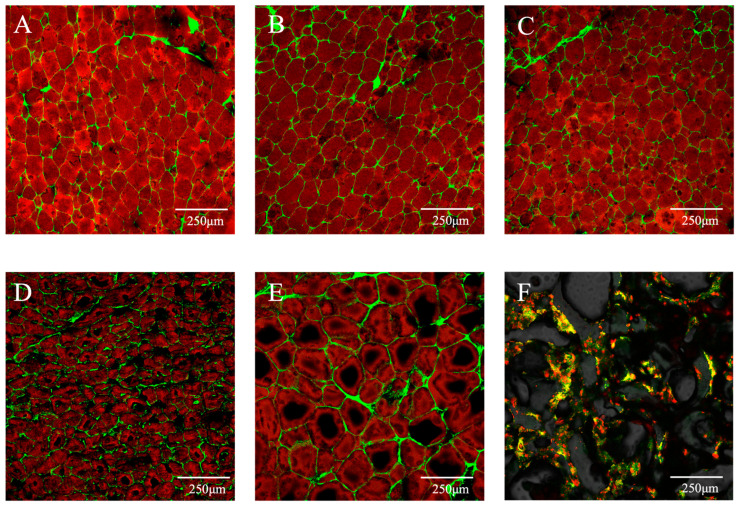
Microstructure of animal meat adipose tissue from different sources under laser confo calmicroscopy. (**A**) Pork back fat; (**B**) pork bacon fat; (**C**) pork lard; (**D**) beef brisket fat; (**E**) beef plate fat; (**F**) fat analogues. The red part is fat, and the green part is protein and polysaccharides.

**Figure 7 foods-14-04355-f007:**
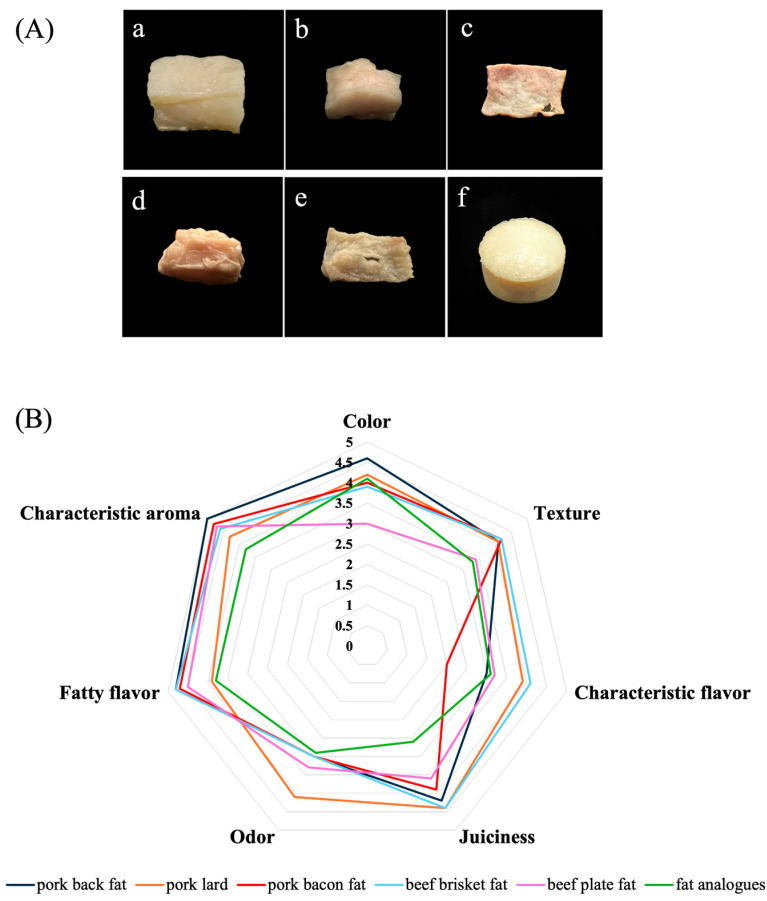
Photographic appearance and aroma profiles of fat analogues. (**A**) Appearance of different fat and fat analogues after cooking: (**a**) pork back fat, (**b**) pork lard, (**c**) pork bacon fat, (**d**) beef brisket fat, (**e**) beef plate fat, (**f**) fat analogues. (**B**) Sensory evaluation profiles.

## Data Availability

The original contributions presented in the study are included in the article. Further inquiries can be directed to the corresponding author.

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
