# Peer review of "Construction and Textural Properties of Plant-Based Fat Analogues Based on a Soy Protein Isolate/Sodium Alginate Complex Coacervation System"

_foods, 2025, doi:10.3390/foods14244355_

Round 1
Reviewer 1 Report
Comments and Suggestions for Authors
Dear editor and authors
This study offers an innovative and successful method for producing vegetable fat analogs with promising microstructural and textural results, demonstrating clear potential for use in meat alternatives. However, its applicability remains limited by the absence of tests in real food systems, a small sensory panel, and a lack of data on storage stability, nutritional implications, and industrial scalability. Despite these gaps, the work provides a solid foundation for future developments.
Some suggestions and corrections are shown below.
Introduction
- Only theses proteins: e.g., gelatin, casein, or whey protein
- All iteractions herer are noncovalent?
non-covalent interactions such as hydrogen bonding, π–π stacking, van der Waals forces, electrostatic interactions, and dipole interactions
- It is necessary add reference here: The quality of existing fat analogues remains inferior to that of traditional animal fat, largely because they lack a microstructure resembling adipose tissue and insufficiently tolerate external forces and temperature fluctuations. Animal fat tissue is composed of a hydrated connective tissue matrix embedded with adipocytes ranging from 50 to 200 μm in diameter.
- The introduction lacks references.
Materials and Methods
- Milli-Q or deionized water was used as described in item 2.1.
- Adding a figure showing the schematic of how the oil-filled microcapsules were prepared would be interesting.
- replace ml by mL
- The curdlan (set as 3% (w/v), 4% (w/v) and 5% (w/v), respectively?
- The mass of the steamed sample was weighed and recorded as M1 and the raw sample was M2 to calculate the steaming loss according to the equation (3): and where M1 was the weight of the sample before steaming, g; M2 was the weight of the sample after steaming, g.
- Use min or minute in all text
- A compression test was conducted according to the method described by Wang, using ª – it is necessary add reference number.
- It is necessary to standardize the way of writing:It is necessary to standardize the way of writing: see: 10 × 10 × 10 mm and 30 mm × 30 mm× 30 mm
- w/v or m/v?
- Explain it 0.1% m/v in 30% acetone
- define FITC
- TCS SP8 confocal laser scanning microscope (CLSM) equipped with an HC PL APO 10x/0.4 CS objective - add manufacturer information.
Results and Discussion
- Figures 1-6 are presented before the text. The text should be presented first, followed by the figure.
- During complex coacervation, when the pH is significantly lower than the isoelectric point of the protein, the positively charged SPI and negatively charged SA interact electrostatically. Explain Figure 2.
Conclusion
- The conclusion is too long and should be shortened.
References
- The references are not formatted the same way; they have at least two different formatting styles.

Dear editor and authors
This study offers an innovative and successful method for producing vegetable fat analogs with promising microstructural and textural results, demonstrating clear potential for use in meat alternatives. However, its applicability remains limited by the absence of tests in real food systems, a small sensory panel, and a lack of data on storage stability, nutritional implications, and industrial scalability. Despite these gaps, the work provides a solid foundation for future developments.
Some suggestions and corrections are shown below.
Introduction
- Only theses proteins: e.g., gelatin, casein, or whey protein
- All iteractions herer are noncovalent?
non-covalent interactions such as hydrogen bonding, π–π stacking, van der Waals forces, electrostatic interactions, and dipole interactions
- It is necessary add reference here: The quality of existing fat analogues remains inferior to that of traditional animal fat, largely because they lack a microstructure resembling adipose tissue and insufficiently tolerate external forces and temperature fluctuations. Animal fat tissue is composed of a hydrated connective tissue matrix embedded with adipocytes ranging from 50 to 200 μm in diameter.
- The introduction lacks references.
Materials and Methods
- Milli-Q or deionized water was used as described in item 2.1.
- Adding a figure showing the schematic of how the oil-filled microcapsules were prepared would be interesting.
- replace ml by mL
- The curdlan (set as 3% (w/v), 4% (w/v) and 5% (w/v), respectively?
- The mass of the steamed sample was weighed and recorded as M1 and the raw sample was M2 to calculate the steaming loss according to the equation (3): and where M1 was the weight of the sample before steaming, g; M2 was the weight of the sample after steaming, g.
- Use min or minute in all text
- A compression test was conducted according to the method described by Wang, using ª – it is necessary add reference number.
- It is necessary to standardize the way of writing:It is necessary to standardize the way of writing: see: 10 × 10 × 10 mm and 30 mm × 30 mm× 30 mm
- w/v or m/v?
- Explain it 0.1% m/v in 30% acetone
- define FITC
- TCS SP8 confocal laser scanning microscope (CLSM) equipped with an HC PL APO 10x/0.4 CS objective - add manufacturer information.
Results and Discussion
- Figures 1-6 are presented before the text. The text should be presented first, followed by the figure.
- During complex coacervation, when the pH is significantly lower than the isoelectric point of the protein, the positively charged SPI and negatively charged SA interact electrostatically. Explain Figure 2.
Conclusion
- The conclusion is too long and should be shortened.
References
- The references are not formatted the same way; they have at least two different formatting styles.
Reviewer 2 Report
Comments and Suggestions for Authors
The paper presents a detailed investigation into the development of plant-based fat analogues through a complex coacervation method that utilizes soy protein isolate (SPI) and sodium alginate (SA). The study emphasizes the potential of microcapsules to mimic animal adipose tissue and thereby improve the texture and culinary qualities of meat products. It is an interesting work and close to the research trends in finding new plant based alternatives.
Introduction: While the paper references relevant studies, integrating a more extensive comparison with existing fat substitutes could enrich the discussion on effectiveness and innovation. For instance, the authors can exploit the work “Phase separation and formation of sodium caseinate/pectin complex coacervates: effects of pH on the complexation” in their for improvement of the study.
Materials and methods: Data Graphical representation of results could be enhanced. For example, clearer charts showing the textural comparisons and sensory evaluation could provide stronger visual support for the findings.
Conclusion: The conclusion could benefit from a discussion on potential applications beyond meat products. Exploring implications for other food systems, such as dairy or bakery, could broaden the impact of the research.
minor comemnts:
Line 96-106: what’s the total biopolymer concentration?
What’s the pH of coacervation? Please check the work ‘Characterization of caseinate-pectin complex coacervates as a carrier for delivery and controlled-release of saffron extract” and use in your study?
Line 155-165: texture analysis: please see the proper literature in the journal of texture studies to clearly show the details for example Textural properties of alginate‐guar gels. I hope this comments will improve the paper.
Round 2
Reviewer 2 Report
Comments and Suggestions for Authors
The comments are replied properly.
Author Response
Thank you for your review and valuable suggestions.